# Changes in visual attentional behavior in complex regional pain syndrome: A preliminary study

Yukiko Shiro[1,2], Shuhei Nagai[1‡], Kazuhiro Hayashi[1‡], Shuichi Aono[3‡], Makoto Nishihara[1‡], Takahiro Ushida[1]*

1 Multidisciplinary Pain Center, Aichi Medical University, Nagakute, Aichi, Japan, 2 Faculty of Rehabilitation Sciences, Department of Physical Therapy, Nagoya Gakuin University, Nagoya, Aichi, Japan, 3 Department of Pain Data Management, Aichi Medical University, Nagakute, Aichi, Japan

These authors contributed equally to this work.
‡ These authors also contributed equally to this work
* ushidat@aichi-med-u.ac.jp

## Abstract

### Purpose

The purpose of the present study was to investigate the visual attentional behavior towards a pain-affected area and face/body images using eye tracking in complex regional pain syndrome (CRPS) patients. Moreover, we investigated the relationship between visual attentional behavior and clinical symptoms.

### Patients and methods

Eight female patients with CRPS type 1 in their upper limbs and 8 healthy adult women participated in this study. First, the participants were asked to watch videoclips in a relaxed manner (Videoclip 1 featured young adults who introduced themselves; Videoclip 2 featured young adults touching the hand of the other person sitting across from them with their hand.) Eye movement data were tracked with eye-tracking glasses.

### Results

In video clip 1, the fixation duration (FD) and fixation count (FC) on faces tended to be lower in CRPS patients than in healthy controls. This tendency was found in patients with low body cognitive distortions. In video clip 2, CRPS patients displayed significantly lower FD and FC on the unaffected hand while watching a video of the unaffected hand being touched compared with healthy controls. Moreover, patients with low body cognitive distortion displayed significantly longer FD on the affected hand.

### Conclusion

Some CRPS patients differed in visual attentional behavior toward the face and body compared with healthy controls. In addition, our findings suggest that patients with lower body

**Data Availability Statement:** The data underlying the results presented in the study are available at https://figshare.com/articles/dataset/Changes_in_

visual_attentional_behavior_in_complex_regional_
pain_syndrome_A_preliminary_study/13258589

**Funding:** The authors received no specific funding
for this work.

**Competing interests:** The authors have declared
that no competing interests exist.

cognitive distortion may have a high visual attention for the affected hand, while patients
with higher distortion may be neglecting the affected hand.

## Introduction

Patients suffering from complex regional pain syndrome (CRPS) display symptoms of refractory pain such as allodynia and hyperalgesia, sensory, motor, trophic and autonomic symptoms in one limb. It is thought that these symptoms are caused and modulated by changes in the peripheral nervous system and plastic changes in the central nervous system [1]. In addition to severe pain, other clinical features include body schema abnormalities (i.e. body perception disturbance) [2], a decreased attention on the painful limb and decreased use in movement [3].

Several studies have shown that patients with CRPS change their attention to the affected limb, relative to the unaffected limb and its surrounding space [4–6], and these changes are commonly referred to as "neglect-like symptoms (NLS)". For example, Bultitude JH, et al. reported that CRPS patients processed visual stimuli more slowly on the affected side, and there was a bias of visual attention away from the affected hand [4]. Filbrich L, et al. reported a visuospatial defect in the immediate vicinity of the affected limb in CRPS patients [5]. Since the previous study showed that the severity of NLS correlates with symptoms of psychological disorders [7], there are concerns about the effect of NLS on psychological status in CRPS patients. Therefore, we thought that it is clinically significant to clarify the pathogenesis of NLS.

However, Christophe L. (2016) [8] and Halicka M. (2020) [9] reported that not all patients with chronic CRPS exhibit decreased spatial attention on the affected side. Moreover, there were reports that the spatial biases shifted towards the affected side [10], which was inconsistent. Furthermore, most studies of spatial bias in CRPS have assessed responses to the task such as temporal order judgment (TOJ) tasks [4,5,9,11], and there have been no reports that have directly assessed overt visual exploration in patients.

Recently, an eye-tracking system has been utilized to study underlying visual attention abnormalities in body perception disturbance (e.g. dysmorphic disorder) [12], psychological disorders (e.g. depression, and autism spectrum) [13,14] and chronic pain (e.g. daily headache, slight musculoskeletal pain) [15,16] and has discerned spatial attention biases in these disorders. For example, a person with body dysmorphic disorder (BDD) paid more attention (i.e. increased fixations and dwell time) to their body parts that they thought were unattractive [12]. Depressed patients are more likely to gaze at sad faces than happy ones, compared with non-depressed subjects [17]. Chronic headache participants demonstrated a significantly greater visual attentional bias (i.e. fixation duration and the location of the initial shift in gaze) toward pain-related information (painful face images) even when other emotional stimuli (e.g. angry, happy, and neutral face images) were presented [15]. More specifically, it is thought that visual attention is influenced not only by pain-related information but also by other physical and psychological information. In addition, these patients had increased visual attention to information related to their symptoms. Thus, we also imagine that CRPS patients would have not only changes in spatial attention, but also different visual attentional behavior compared with healthy individuals.

The present study was to investigate visual attentional behavior while watching the video of others, face-to-face, in order to identify whether CRPS patients have a visual attentional bias toward the bodies of others in an external space (video 1). Secondly, to identify visual

attentional biases towards the affected area in CRPS patients, we examined visual attentional behavior while watching a video in which a stranger seemed to be touching a patient's hands face-to-face (video 2). In addition, we investigated the relationship between visual attentional behavior and clinical symptoms, including psychological states.

## Materials and methods

### Participants

Eight female CRPS patients who displayed symptoms and signs in one upper limb (age, mean, 45.0 ± 10.6 standard deviation years, Table 1) and eight age-matched healthy adult females (mean, 43.2 ± 13.1 standard deviation years) were enrolled in this study. Both patients and healthy subjects were right-handed. The clinical symptoms of all CRPS patients met the Japanese CRPS diagnostic criteria for clinical purposes [18] without a neurological deficit in the limbs, and all of them were diagnosed with CRPS type 1 by their attending physicians. In addition, the team conference in the multidisciplinary pain center ensured that the patient met the diagnostic criteria for CRPS. The disease duration of those patients was 19.3 ± 10.1 months. All patients were treated as outpatients at the Multidisciplinary Pain Center of Aichi Medical University Hospital. Control participants were recruited through flyers posted on a notice board. All participants provided written informed consent prior to participation in this study. This study was approved by the ethical committee of Aichi Medical University (reference number: 15-H079). Moreover, this study was conducted in accordance with the Declaration of Helsinki.

### Pain and other clinical evaluations

Numerical rating scale (NRS) scores (in which '0' indicates 'no pain' and '10' represents 'the greatest pain possible') were used to obtain the average severity of total pain over a period of one week.

Disability was assessed by the Pain Disability Assessment Scale (PDAS). Scores for the total PDAS range from 0 to 60, with higher scores indicating higher levels of pain interference. The cut-off point for distinguishing between patients with and without chronic pain was 10 points

**Table 1. Characteristics of CRPS patients.**

| Patient No. | Age (year) | Painful side | Pain severity (NRS) | Duration of symptoms (month) | PDAS | PCS | HADS anxiety | HADS depression | BPDS | Inciting event |
|---|---|---|---|---|---|---|---|---|---|---|
| 1 | 35 | Left | 9 | 21 | 45 | 28 | 8 | 10 | 17 | Blood sampling |
| 2 | 45 | Right | 8 | 40 | 39 | 32 | 9 | 13 | 15 | Traffic accident (whiplash) |
| 3 | 53 | Right | 8 | 23 | 40 | 32 | 8 | 17 | 36 | Traffic accident (contusion of upper limb) |
| 4 | 44 | Left | 7 | 81 | 39 | 14 | 3 | 10 | 44 | Traffic accident (whiplash) |
| 5 | 52 | Left | 7 | 26 | 22 | 35 | 6 | 13 | 36 | Nonspecific severe neck shoulder pain* |
| 6 | 63 | Right | 7 | 13 | 43 | 32 | 9 | 13 | 21 | After surgery (syndesmoplasty of wrist) |
| 7 | 34 | Right | 6 | 17 | 39 | 25 | 3 | 6 | 38 | Traffic accident (contusion of upper limb) |
| 8 | 34 | Right | 6 | 9 | 34 | 43 | 10 | 9 | 42 | Blood sampling |

BPDS: Bath CRPS body perception disturbance scale, CRPS: Complex regional pain syndrome, HADS: hospital anxiety and depression scale, NRS: Numerical rating scale, PCS: Pain catastrophizing scale, PDAS: Pain disability assessment scale, PSEQ: Pain self-efficacy scale.

* Non-specific severe neck pain: She had a long history of stiff neck and shoulders, after slept wrong, she developed severe non-specific neck pain with no major neurological/ radiological findings, developed CRPS symptoms afterwards.

in the Japanese version of the PDAS validity study [19]. Moreover, the pain catastrophizing scale (PCS) was used to assess catastrophic thinking related to pain, and the hospital anxiety and depression scale (HADS) was used to assess anxiety and depression. The total PCS score can range from 0 to 52, with higher scores indicating higher levels of catastrophizing. A PCS score of more than 30 corresponds to the 75th percentile of the distribution of PCS scores in clinic samples of chronic pain patients. The mean value of the PCS score among Japanese patients with chronic low back pain was 35.1 [20]. The HADS score consists of 14 items; the anxiety and depression subscales each include 7 items, with possible scores for each subscale ranging from 0 to 21. In the reliability and validity study for the Japanese version of HADS, the optimal cut-off point for screening for adjustment and major depressive disorders was an anxiety score of 10, and a depression score of 11, with sufficient sensitivity and specificity (91.5 and 65.4%, respectively) [21]. The body schema abnormality was assessed by the Bath CRPS Body Perception Disturbance Scale (BPDS). The BPDS provides an assessment of the extent to which body perception disturbance is experienced, and a higher score denotes greater disturbance, with 57 being the maximum total score [22].

## Experimental settings

All participants were seated in front of a 32-inch TFT (Thin Film Transistor) flat screen monitor, wearing a head-mounted eye-tracker. The monitor was placed 70 cm away from the participants. Participants' eye movement data were tracked with eye-tracking glasses (Tobii Pro Glasses 2, TobiiTechnology, Danderyd, Sweden). Eye-tracking glasses were equipped with two cameras for each eye, using near-infrared illumination to create reflection patterns on the camera and pupil. The sampling frequency in the eye tracker was 50 Hz. A nine-point calibration was conducted prior to each measurement to ensure good quality of the eye-gaze recordings. Participants were instructed to watch the screen, and were not allowed to look elsewhere during the recording [23]. In addition, before starting the main testing, they were instructed to focus their gaze on the cross marks and not to move their gaze for calibration. Then the participants were asked to look at two types of silent video clips (video 1 for experiment 1, video 2 for experiment 2) in a relaxed manner.

## Video 1

After the cross marks were shown to the participants (3 sec.), test subjects were instructed to watch the silent video clip 1 in which nine young persons (5 men and 4 women) introduced themselves through the display (Fig 1). During this session, their eye movements were tracked/measured and Fixation Count (FC) and Fixation Duration (FD) were analyzed using the eye-tracking system.

Fixation Count (FC): The number of times a participant's gaze fixed on the areas of Interest (AOI; defined below) such as face (the dotted area in Fig 1), affected side of the forearm, non-affected side of the forearm (forearms include hands, the shaded area in Fig 1), body area (the horizontal line area in Fig 1). Fixation Duration (FD): The total duration of a participant's gaze on the AOI such as face, arm, or other body area during the video clip.

## Video 2

Video clip 2, a silent video clip, featured young adults (Person A) touching the hand of Person B sitting across from them with their right hand (7 seconds long × 8 times; one man and one woman touched the right and left hand four times each, Fig 1). The young adults in video clip 2, including person B, were the same person as in video clip 1.

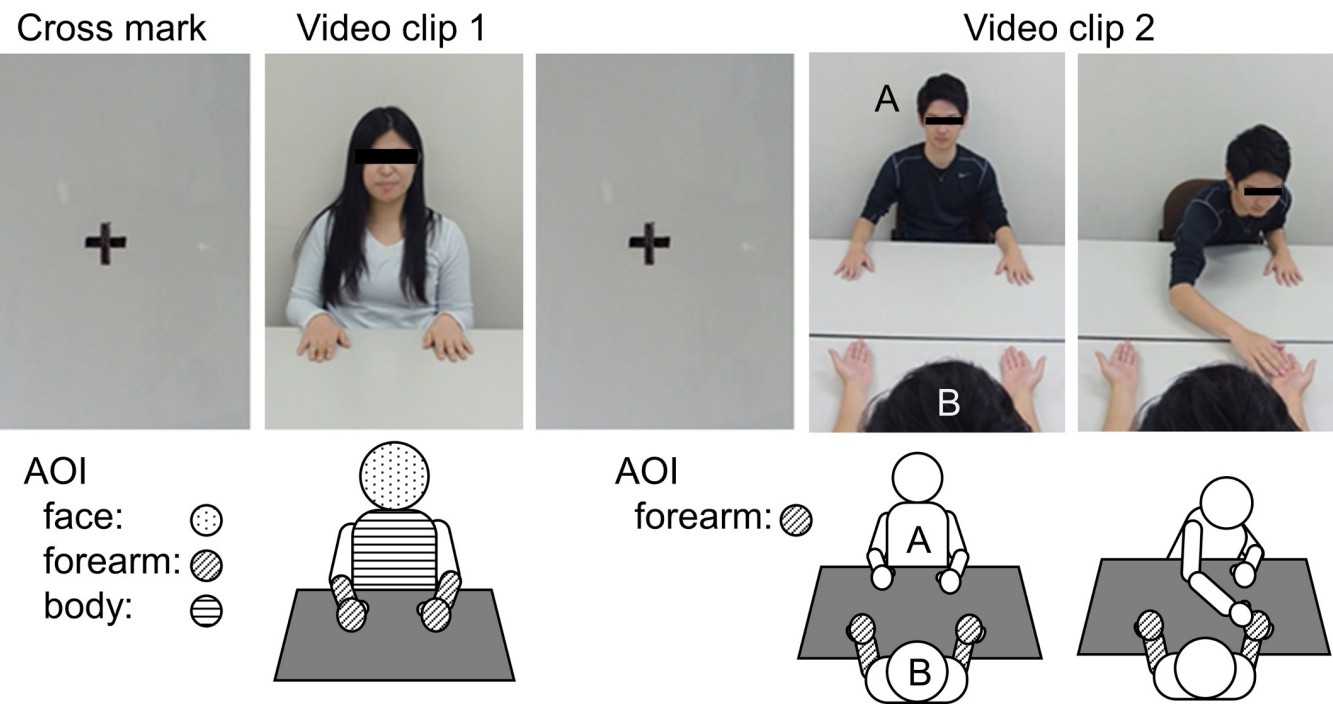

**Fig 1. Video clips 1 and 2.** In video clip 1, a video (without audio) of a young person talking was presented for 7 sec. A total of nine videos of the young person (5 men and 4 women) were presented, and the cross-mark was presented for 3 sec. during each video. In video clip 2, a video (without audio) of Person A touching the hand of Person B was presented for 7 sec. The role of Person A was performed by one man and one woman. A video of Person A touching the right and left hand of Person B was presented 4 times (two times each for man and woman). In addition, cross-marks were presented for 3 sec. during each video. The illustration below shows the AOI for each video clip. AOI: areas of Interest.

The FC and FD were measured and analyzed at each site (painful side arm, non-painful side arm of person B, the shaded area in Fig 1) when the video clip of the painful or non-painful arm being touched was played on the monitor. The same analytical paradigms were followed with the control group as well.

Also, the young adults who appeared in Videos 1 and 2, all of whom were Asian college students, did not know any of the participants. Furthermore, no scars or specific deformities were seen on their hands. In addition, informed consent was obtained from the young adults in the image.

### Details of eye movement analysis (fixation count and fixation duration)

Eye-movement data were analyzed using analysis software (Tobii Pro Lab, Tobii Technology, Danderyd, Sweden) [23,24]. The AOI accurately specified areas on each video clip as follows:

Video clip 1; face, body, and both forearms of the speaking person, Video clip 2; right and left hands of person B (Fig 1).

The parameter from the eye tracking was the total FC and FD. The Tobii Pro Studio used the Velocity-Threshold Identification (I-VT) fixation classification algorithm to identify the fixation [24]. The fixation is defined as the eye-movement data point that is below the velocity threshold. In this study, a fixation was identified as when the mean horizontal and vertical eye position co-ordinates sustained eye movement at a location within 30°/s of the visual angle [24]. Duration was the total fixation time within each AOI. The total count was the number of fixations within each AOI.

## Statistical analysis

Normality of distribution for each measurement was evaluated using the Shapiro-Wilk test for continuous variables. The statistical differences between the groups were analyzed using the Mann-Whitney U test. Comparisons of the groups were analyzed using the Wilcoxon matched pairs test. In addition, one-way repeated measures ANOVA followed by Bonferroni post hoc comparisons were also used to compare each AOI of video clip 1, and the Kruskal-Wallis test followed by Steel-Dwass post hoc comparisons were used to compare each FD/FC of the touched side or contralateral side hand in CRPS and healthy controls in video clip 2. An effect size (r) was calculated to determine the magnitude of difference between the groups. Effect size (r) of $\geqq 0$ and $< 0.1$ was classified as 'no effect'; $\geqq 0.1$ and $< 0.3$ as 'a small effect'; $\geqq 0.3$ and $< 0.5$ as 'a moderate effect'; and $\geqq 0.5$ as 'a large effect'. The relationships between FD/FC of an AOI and clinical evaluations were analyzed using Spearman correlation coefficients. All data were analyzed using JMP software (version 14, SAS Institute, Cary, NC, USA), and the level of significance was set to 5%.

## Results

CRPS patients had moderate to high intensity pain (NRS 6–9). Moreover, the PDAS was above the cut-off score in all CRPS patients. Only two patients had a PCS score of 35 or higher, the mean score for Japanese chronic pain patients [20]. There was also only one patient with HADS-anxiety above the cut-off score [21] However, HADS-depression was above the cut-off for half of the patients [21]. The BPDS ranged from 17 to 44 points and included patients with severe and mild body schema abnormality (Table 1).

### Video 1

There was no significant difference in FDs or FCs on all AOIs between CRPS patients and healthy controls (Table 2). However, in healthy controls, facial FDs and FCs were significantly higher than other body parts (p <0.05, Table 2). In contrast, facial FDs were significantly longer in CRPS patients than FDs on both forearms, and facial FCs were significantly higher than non-painful forearm FCs (p< 0.05, Table 2), but there was no significant difference between the face and body in FDs or FCs (p = 0.664, 1.00, Table 2). Two of the CRPS patients had markedly few facial FDs and FCs and/ or high FDs and FCs of the body (Fig 2). The painful and non-painful sides refer to the sides of the person in the video corresponding to the painful (VP) and non-painful (VNP) sides of the patients.

**Table 2. Eye movement data of video clip 1.**

| Fixation parameter | AOI | CRPS patients | | Healthy controls | | p value | | effect size (r) | |
|---|---|---|---|---|---|---|---|---|---|
| | | Painful side | Nonpainful side | Right | Left | | | | |
| Fixation duration (sec) | Face | 18.46±17.90* | | 30.88±7.62* | | 0.161 | | 0.520 | |
| | Body | 11.71±15.24 | | 4.58±3.69* | | 0.721 | | 0.148 | |
| | Forearm | 1.21±1.32 | 0.28±0.56* | 0.93±0.99 | 0.59±0.88* | 1.000 | 0.388 | 0.000 | 0.305 |
| Fixation count | Face | 29.63±27.95# | | 59.50±23.31* | | 0.065 | | 0.651 | |
| | Body | 29.50±29.13 | | 20.88±16.25* | | 0.878 | | 0.074 | |
| | Forearm | 5.00±4.11 | 1.75±3.41# | 2.88±2.70 | 2.75±3.88 | 0.339 | 0.506 | 0.338 | 0.235 |

Value: Mean ± SD, AOI: Area of interest, CRPS: Complex regional pain syndrome.

p value: compared between CRPS patients and Healthy controls. Forearm compared with the painful side and the right side, the nonpainful side and the left side.

*, vs. Body or Forearm (both right and left): p<0.05,

#, vs. Nonpainful side forearm: p<0.05.

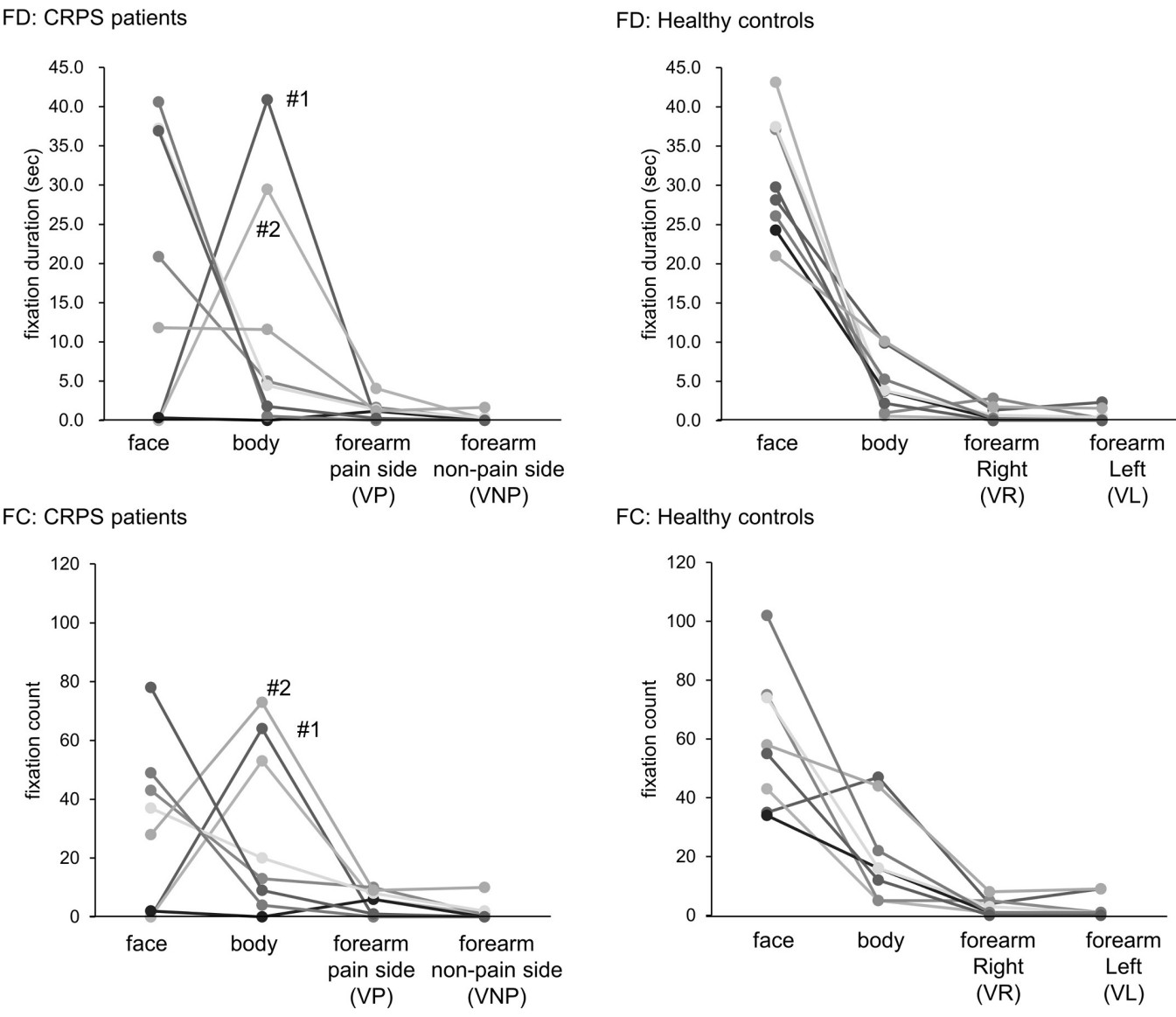

**Fig 2. Individual data of visual attentional behavior while watching video clip 1.** FD: fixation duration, FC: fixation count, VP and VNP: VP and VNP refer to the side of the person in video 1 corresponding to the painful and non-painful side of the patients. VR and VL: VR and VL refer to the right and left side of the person in video 1.

Fig 3 shows a comparison of VP and VNP sides, the left and right sides of the forearm in CRPS patients and healthy controls. The left and right sides refer to the left (VL) and right (VR) sides of the person in the video. There were no significant differences in both FDs and FCs. However, there was a higher tendency of FCs on the VP side of the forearm compared with the VNP side in CRPS patients (p = 0.058, effect size = 0.389).

BPDS and FDs on the body showed a significant negative correlation (Fig 4B). In contrast, FDs of the face and forearm did not correlate with BPDS (Fig 4A, 4C and 4D). Namely, patients with low body cognitive distortions (low BPDS) had longer FDs on the body. Other clinical parameters, such as pain intensity, disability, disease duration, and psychological state (PCS, HADS) did not show significant correlations with the eye-movement data.

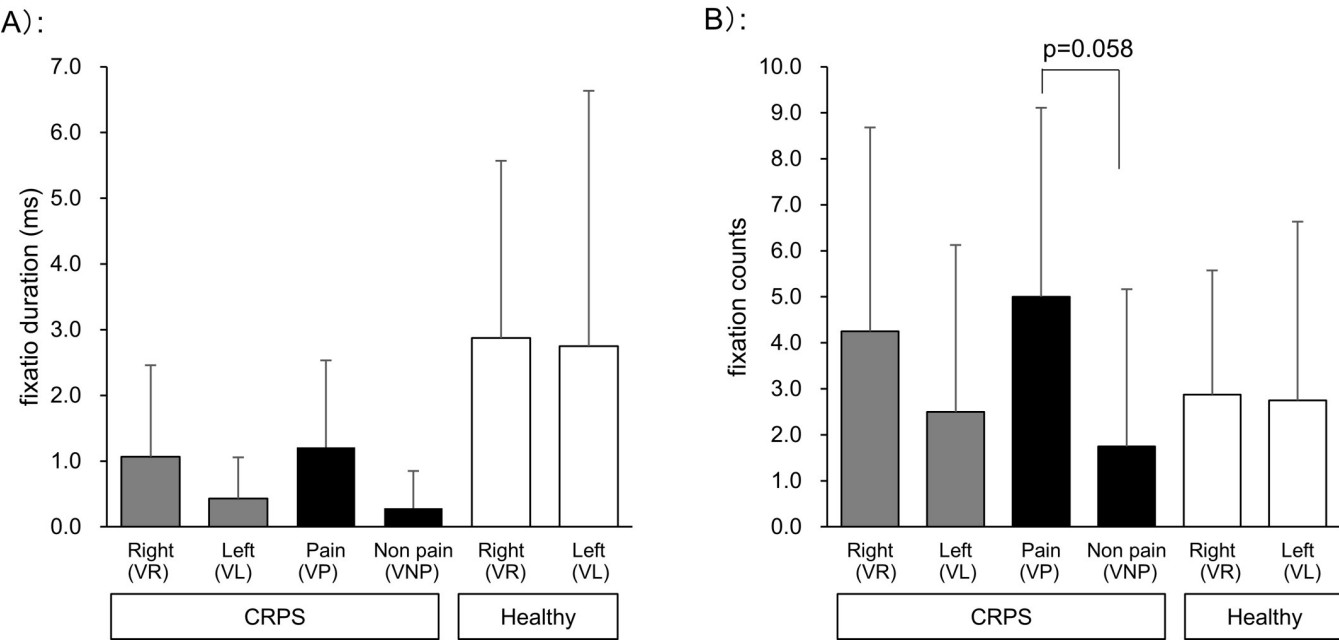

**Fig 3. Visual attentional behavior to forearm in video clip 1: Comparison of left and right sides or the painful side and the non-painful side.** A) Fixation duration, B) Fixation count. Value: mean and standard deviation. CRPS: complex regional pain syndrome, VP and VNP: VP and VNP refer to the side of the person in video1 corresponding to the painful and non-painful side of the patients. VR and VL: VR and VL refer to the right and left side of the person in video 1.

### Video 2

In the control group, both FDs and FCs of the right and left hands of person B on the monitor were significantly higher on the touched side than on the contralateral hand. (Table 3).

In contrast, there were no significant differences in either FDs and FCs between the touched side and contralateral side hand in the CRPS group. However, there was a higher tendency for FDs on contralateral hands (painful side) compared with the touched side hand (non-painful side) while watching the non-painful touch video (p = 0.068, effect size = 0.646, Table 3). The painful side and non-painful sides refer to the side of person B in video 2 (Fig 1) corresponding to the painful and non-painful sides of the patients. Incidentally, there was no difference between the left and right in the CRPS group (Table 3).

Moreover, FDs on the non-painful side while watching the non-painful side hand touch video in the CRPS group were shorter than FDs in the control group (vs. right / left hand of healthy controls: p = 0.019/ 0.131, effect size = 1.031/ 0.768, Fig 5A). There was also a lower tendency for FCs in the CRPS group compared with the control group (vs. right / left hand of healthy controls: p = 0.067/ 0.294, effect size = 0.868/ 0.622, Fig 5C). Comparisons between ipsilateral hands (right vs. right and left vs. left) showed that the FDs and FCs on the left hand while watching the left hand touch video tended to be lower in the CRPS group than in healthy controls (p = 0.141, 0.338, effect size = 0.765, 0.591, Fig 5B and 5D). The right and left hands refer to the right and left hands of person B in video 2 (Fig 1). In contrast, FDs and FCs on the contralateral side hand did not show a clear difference between the groups.

In addition, BPDS and FDs on the painful side hand showed a significant positive correlation, both when touching the painful side and when touching the non-painful side hand (Fig 4E and 4F).

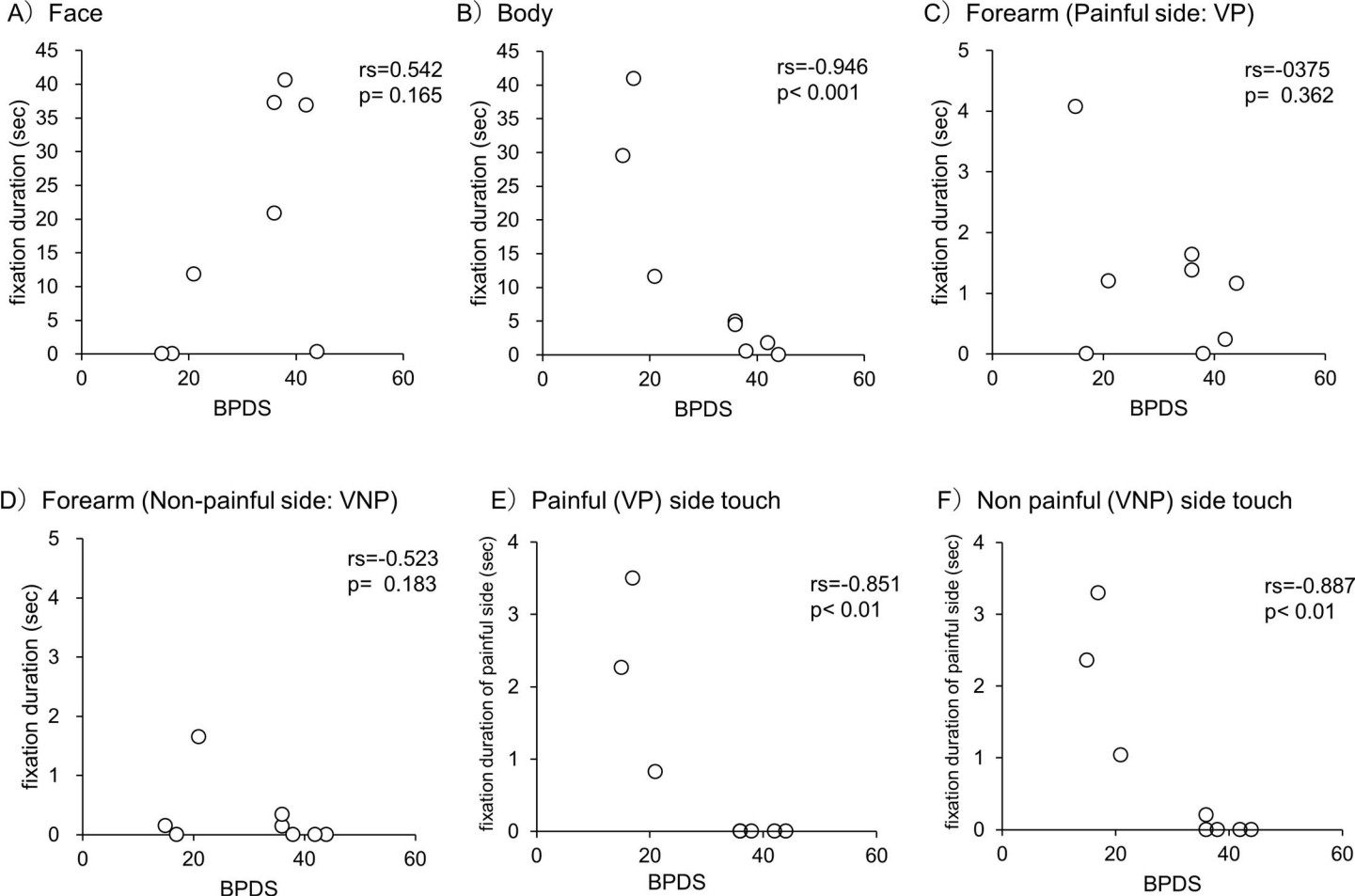

**Fig 4. Correlation with BPDS and fixation duration.** A-D): video clip 1, E, F): video clip 2. BPDS: Bath CRPS Body Perception Disturbance Scale, VP and VNP: VP and VNP refer to the side of the person in video 1 corresponding to the painful and non-painful side of the patients. VR and VL: VR and VL refer to the right and left side of the person in video 1.

As with video 1, other clinical parameters did not show any association with the eye-movement data.

## Case reports

**Case 1** (Table 1; No.1, Fig 6A and 6B, painful side: left).

This patient with low body cognitive distortions (low score of BPDS) looked at the trunk (high fixation on the body), not the face, while patients introduced themselves through the display (while watching video clip 1) in video 1. In video 2, she stared at the painful side, even before it was touched by a hand. Note that healthy controls gazed at the face in video 1 (Fig 6E) and did not gaze at the hand before it was touched in video 2 (Fig 6F).

**Case 2** (Table 1; No.4, Fig 6C and 6D, painful side: right).

We observed this patient with high body cognitive distortions (high BPDS) ignoring the subject during the self-introduction through the display (while watching video clip 1) and when touching the painful side in both videos 1 and 2. Note that healthy controls gazed at the face in video 1 (Fig 6E) and at the hands being touched in video 2 (Fig 6F).

**Table 3. Eye movement data of video clip 2.**

| Fixation parameter | Subject | Touch side | AOI | | p value | effect size (r) |
|---|---|---|---|---|---|---|
| | | | Touch side hand | Contralateral hand | | |
| Fixation duration (sec) | CRPS | Painful side | 0.82±1.34 | 0.00±0.00 | 0.109 | 0.567 |
| | | Nonpainful side | 0.02±0.06 | 0.86±1.29 | 0.068 | 0.646 |
| | | Right side | 0.38±0.81 | 0.44±1.56 | 1.000 | 0.000 |
| | | Left side | 046±1.23 | 0.42±0.86 | 1.000 | 0.000 |
| | Control | Right side | 0.99±1.16 | 0.0±0.0 | 0.043 | 0.715 |
| | | Left side | 1.81±2.74 | 0.0±0.0 | 0.018 | 0.837 |
| Fixation count | CRPS | Painful side | 0.75±1.16 | 0.00±0.00 | 0.109 | 0.567 |
| | | Nonpainful side | 0.25±0.71 | 0.88±1.23 | 0.197 | 0.456 |
| | | Right side | 0.38±0.74 | 0.38±0.74 | 1.000 | 0.000 |
| | | Left side | 0.63±1.19 | 0.50±1.07 | 0.785 | 0.096 |
| | Control | Right side | 1.13±1.36 | 0.0±0.0 | 0.039 | 0.728 |
| | | Left side | 1.63±1.85 | 0.0±0.0 | 0.014 | 0.868 |

Value: mean ± SD, AOI: Area of interest, CRPS: Complex regional pain syndrome, *p* value: Touched side hand vs. contralateral hand.

## Discussion

### Visual attention behaviors towards the faces of others

In the present study, we investigated visual attentional behavior using eye-tracking systems in CRPS patients, when they were watching another person speaking on the monitor (video clip 1).

In general, it is known that both healthy children and adults fixate their gaze on the face of speakers until the end of their speech, because facial expressions are a major source of information about the reaction of others in a social interaction [25,26]. However, this study showed that the visual attention on faces tended to be lower in CRPS patients than in healthy controls. In particular, this tendency was found in patients with low body cognitive distortions. On the other hand, it was reported that visual attentional bias for faces is reduced in depressed individuals [27], this trend correlated with the severity of depressive symptoms [28]. However, there was no correlation between HADS-depression and visual attention for the face in the present study.

The previous study reported that CRPS patients show attention bias away from both the affected limb and the affected space [4,5]. Moreover, the degree to which attention was directed away from the affected side is related to BPDS [4]. The results of this study also showed a tendency for decreased attention to the affected limb in patients with high body cognitive distortions. Thus, we thought that the patient with high body cognitive distortions could have gazed at the face as well as healthy controls, because they reduced their attention on the affected limb.

On the other hand, it has also been reported that there was no visuospatial attention bias to the affected space in CRPS patients [8,9]. In addition, it also has been reported that CRPS patients have exaggerated somatosensory input from the affected limb, resulting in an overrepresentation of space on the affected side [19]. The results of the present study showed a positive correlation between BPDS and gazes at the face whereas there was a negative correlation with gazes at the painful forearm. Thus, we thought that the patient with low body cognitive distortions may have decreased visual attention on the face as a result of visual attentional biases to the affected limb. Therefore, the lower visual attention to faces in CRPS patients compared to healthy controls may be due to the inclusion of several CRPS patients with low body cognitive distortions in the subjects. Increased attention to the body may have been due to the attentional biases toward the hands that were closer to the body.

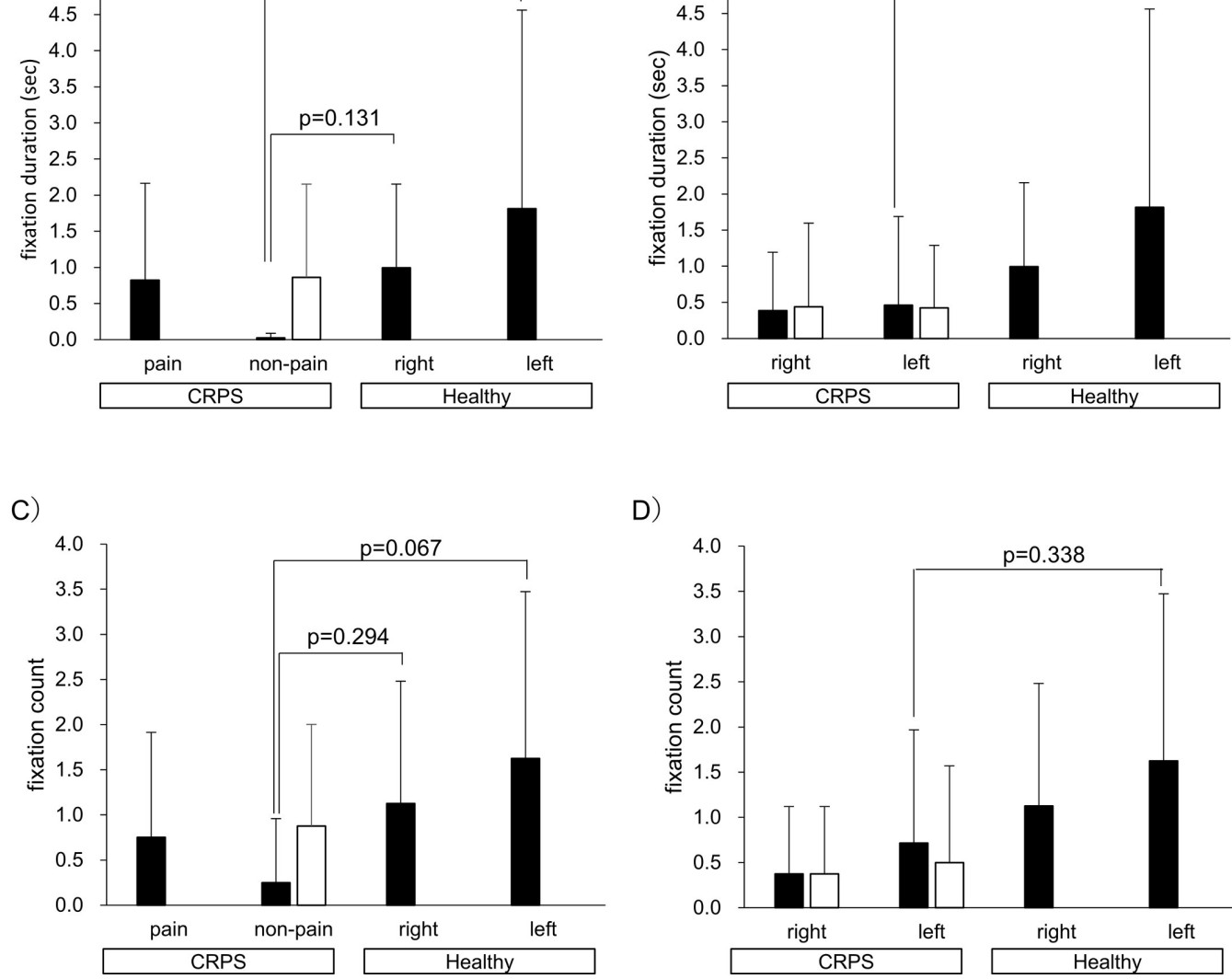

**Fig 5. Fixation duration and count while watching video clip 2: Comparison within the touch side or contralateral side.** A) Fixation duration: Comparison of painful and non-painful side hands in CRPS patients, left and right hands in healthy controls. B) Fixation duration: Comparison of left and right hands in CRPS patients and healthy controls. C) Fixation count: Comparison of painful and non-painful side hands in CRPS patients, left and right hands in healthy controls. D) Fixation count: Comparison of left and right hands in CRPS patients and healthy controls. Value: mean and standard deviation. ■: touched side, □: contralateral side. CRPS: complex regional pain syndrome.

Facial expressions are important for interindividual communication, from which observers can quickly and easily make a number of inferences about identity, gender, age, physical health, attractiveness, emotional state, personality traits, pain or physical pleasure, and even social status [29]. Therefore, we are concerned that reduced visual attention on the face in CRPS patients may lead to social disadvantages.

## Visual attentional biases towards the affected area

CRPS patients showed a tendency for visual attentional bias to the affected side forearm compared with the unaffected side forearm. It was interesting that this visual attentional bias was

## Case 1

### A) Video 1            B) Video 2

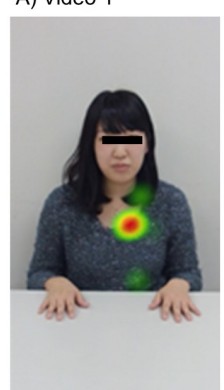 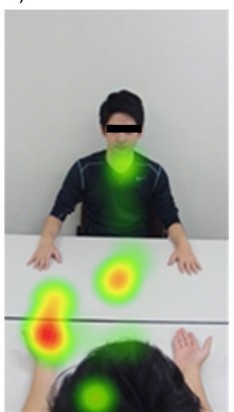

## Case 2

### C) Video 1            D) Video 2

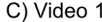

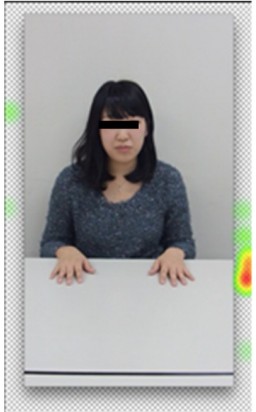 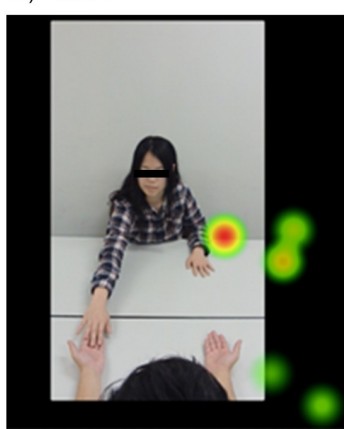

## Healthy control

### E) Video 1            F) Video 2

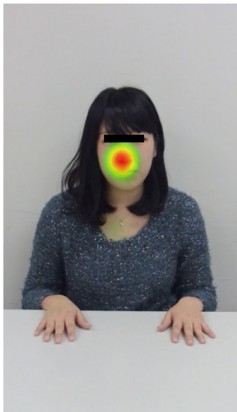 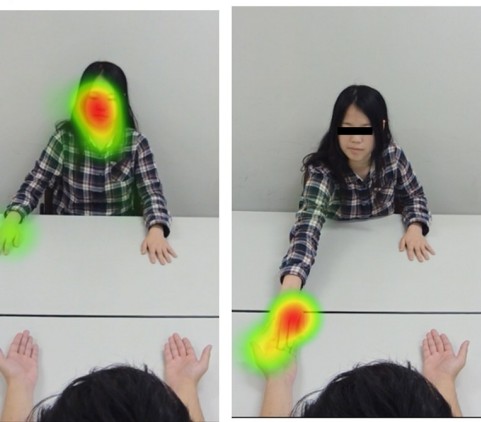

**Fig 6. The heat map of eye behavior performed on Case 1, 2 and healthy controls.** A, B) Case 1 (Table 1; No1,), C, D) Case 2 (Table 1; No.4), E, F) Healthy controls. Areas with a higher percentage of gaze have been indicated in red, and areas with a lower percentage have been displayed in green.

shown for the forearm of others who were face-to-face. (It is not a mirror image of the affected side of patients, but the equivalent affected side of the person in video clip 1). Moreover, in video 2, a few CRPS patients gazed at the affected side hand when watching a video of the unaffected side hand being touched. These patients had low body cognitive distortions. In contrast, in all healthy controls, when watching a video of a hand being touched, they gazed at the touched hand.

It has been suggested that the bias in visual subjective body midline judgments towards the CRPS-affected side is due to an exaggerated somatosensory input from the painful limb [2,6]. For example, Sumitani M. et al. suggested that since the deafferentation of the affected limb caused a transient decrease in pain and a transient shift in the visual subjective body midline deviation toward the unaffected side, the persistent pain state in CRPS distorts visuospatial perception [2]. In fact, two patients who had increased gaze on the affected limb (patient No. 1 and 2) also presented with strong pain (NRS 9 and 8) in the present study. Moreover, studies that have tested attention to explicit visual information about the affected limb found that pain intensity was significantly related to the degree of attentional bias [30]. We speculate that the video of the hand touch used in this study may have provided explicit visual information about the affected limb. However, this hypothesis does not explain the visual attentional bias for the affected forearm of others who were face-to-face (videoclip 1). This was because the affected limb of the person in the video was located on the side corresponding to the space on the unaffected side of the CRPS patients in video clip 1. It was possible that some CRPS patients may have decreased attention to the affected space, regardless of what was displayed on the screen.

## Relationship between body cognitive distortions and visual attention

In contrast, patients with a severe body cognitive distortion neglected the hand on the affected side when they were watching videos of either touch to the affected side or touch to the unaffected side. As previously stated, previous studies have reported that CRPS patients have reduced visual attention to the affected side compared to the unaffected side [4,5]. Especially, the severity of body perception disturbance was found to predict the magnitude of spatial attention bias away from the affected side in people with CRPS [4]. Similar results were found in the present study, in which visual attentional behavior was directly assessed using eye-tracking. Previous studies suggested that altered body representation may interfere with the ability to process information coming from the limb and the space that surrounds it [30,31]. However, such an explanation for the relationship between body representation distortion and attentional bias is negated in CRPS [4]. CRPS patients with severe body perception disturbance commonly reported that the affected limb is psychologically detached from the remainder of their unaffected body in a way that feels alien; a sense of disowning, and something outside of their control [20]. Such phenomena as disownership also often occurred when neglected following brain injury [4,32]. Specifically, these symptoms can follow temporoparietal lesions [6]. The severity of body perception disturbance has been linked to reorganization of the primary and secondary cortical maps of the affected limb in CRPS [33]. Therefore, these body representations and spatial attentional changes may be caused by the cortical changes that occur in CRPS patients.

In this way, there was no consensus as to whether visual attention in CRPS patients deviated to the painful or non-painful side. One possible reason to explain this is that there are different ways to assess visual attentional behavior. However, the results of the present study, which directly assessed visual attentional behavior using eye tracking systems, also showed these two patterns. As suggested by several previous studies [2,4–6,11], we thought that the visual

attentional biases in CRPS patients may be caused by different mechanisms: pain may have a strong influence on the visual attentional biases, and body cognition distortion may have a strong influence on the visual attentional biases.

In general, the incidence and severity of NLS in CRPS increases with longer disease duration and stronger pain intensity [34]. Conversely, it has been reported that non-CRPS limb pain is related to NLS, but there was no relationship between pain intensity and severity of NLS in CRPS [7]. In the results of the present study, the visual attentional behavior of the CRPS patient was not predicted by pain intensity or duration of symptoms, consistent with previous studies that measured spatial attention using TOJ tasks [4,11]. Moreover, it was reported that NLS correlated with psychological factors like anxiety, somatization, and depersonalization in CRPS [7]. There have also been reports that the NLS seen in CRPS is probably the result of a psychological disorder rather than a direct cause of pain [8]. However, the visual attentional behavior did not show an association with only pain intensity but also psychological factors other than body perception disturbance in this study. One of the reasons for the different result was that previous studies assessed NLS using a questionnaire [7]. We thought that the visual attentional behavior of CRPS was more influenced by neuropsychological factors such as body perception disturbance than by psychological factors such as anxiety and depression. However, the absence of significant correlations could be due to the small sample size in this study, and therefore we thought that a study with a larger sample size is needed.

### Limitations and considerations for future research

We should consider the present limitations of this study and future perspectives. First, we did not clarify the mechanisms of the distorted visual attentional behavior in CRPS patients. Secondly, all subjects were women, and the preset sample size used in this study was small. We thought that we need a higher number of subjects and should conduct a more detailed assessment of the severity of individual patients in future studies. In addition, patients suffering from pain other than CRPS were not included in this study. NLS has been reported to occur in patients with other chronic pain as well as CRPS. For example, Frettlöh et al. observed NLS in 80.3% of patients with chronic upper limb pain [35]. Thus, changes in visual attentional behavior may not be specific to CRPS and should be compared with the visual attentional behavior of non-CRPS painful disorder of the upper extremity. Thirdly, there were quite a few different types of inciting events (blood sampling to traffic accident) and this may have influenced the results. Fourthly, the fixation count and fixation duration for specific AOI's were taken as outcome measures in this study. Therefore, we need to consider the gazing behavior towards other areas. Especially, we thought it was necessary to analyze the gaze behavior on the forearm and hand separately. We should also have taken into account individual differences in fixation, such as whether people fixate more or less to begin with. Fifthly, video 1 may have affected the results of video 2, because video 1 was followed by video 2. We need to randomize the order of video 1 and video 2. Sixthly, the visual attentional biases in CRPS patients have been often reported using the TOJ task [4,5,9]. Therefore, we will be able to characterize the visual attentional biases in CRPS patients by examining their visual attentional behavior during the performance of such tasks using eye tracking. Finally, it was not clear whether the changes in visual attentional behavior were a cause or a result of CRPS, because this study was a type of cross-sectional research.

### Conclusion

We investigated visual attentional behavior using eye tracking in CRPS patients. Some patients differed in visual attentional behavior toward the face and body of the speaker compared with

healthy participants. In several patients, visual attention to the faces of others clearly reduced and there was concern that this may lead to social disadvantage. In particular, patients whose body perceptions did not change displayed an increased level of visual attention on the affected hand and a decrease in visual attention on the faces of others. In contrast, patients with body cognitive distortion constantly decreased their visual attention to the affected hand. As such, the visual attentional behavior in CRPS patients was not associated with pain intensity or psychological factors (such as anxiety or depression, catastrophizing). These findings suggest that CRPS patients with less body cognitive distortions may have increased visual attention to the affected limb and decreased visual attention to other spaces and areas.

## Supporting information

**S1 File.**
(XLSX)

## Acknowledgments

The authors would like to thank Matthew McLaughlin for assistance as language editor.

## Author Contributions

**Conceptualization:** Takahiro Ushida.

**Data curation:** Yukiko Shiro.

**Formal analysis:** Yukiko Shiro, Shuichi Aono.

**Investigation:** Yukiko Shiro, Kazuhiro Hayashi.

**Methodology:** Yukiko Shiro.

**Supervision:** Makoto Nishihara, Takahiro Ushida.

**Writing – original draft:** Yukiko Shiro, Shuhei Nagai.

**Writing – review & editing:** Makoto Nishihara, Takahiro Ushida.

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
