## [Decision Letter · Decision Letter 0]

13 Oct 2020

PONE-D-20-25338

Changes in visual attentional behavior in complex regional pain syndrome

PLOS ONE

Dear Dr. Ushida,

Thank you for submitting your manuscript to PLOS ONE. It has been thoroughly reviewed by two experts in the field and, after careful consideration, we feel that it has merit but does not fully meet PLOS ONE’s publication criteria as it currently stands. Therefore, we invite you to submit a revised version of the manuscript that addresses the comments raised during the review process.

Both reviewers have supplied useful lists of points at which more detail could be included. In some places this is to ensure clarity, and to disambiguate content, whereas in other places it to fully elaborate on assertions that have been made in the text. I encourage you to address all of these comments to the best of your abilities, as they will certainly help your submission to maximise its potential impact.

We look forward to receiving your revised manuscript.

Kind regards,

Alastair D. Smith

Academic Editor

PLOS ONE

3.We note that you have indicated that data from this study are available upon request. PLOS only allows data to be available upon request if there are legal or ethical restrictions on sharing data publicly. For information on unacceptable data access restrictions, please see http://journals.plos.org/plosone/s/data-availability#loc-unacceptable-data-access-restrictions.

Reviewers' comments:

Reviewer's Responses to Questions

**Comments to the Author**

1. Is the manuscript technically sound, and do the data support the conclusions?

Reviewer #1: Yes

Reviewer #2: Partly

2. Has the statistical analysis been performed appropriately and rigorously? 

Reviewer #1: Yes

Reviewer #2: Yes

3. Have the authors made all data underlying the findings in their manuscript fully available?

Reviewer #1: No

Reviewer #2: Yes

4. Is the manuscript presented in an intelligible fashion and written in standard English?

Reviewer #1: Yes

Reviewer #2: No

5. Review Comments to the Author

Reviewer #1: This is an interesting study. I have only a few minor comments

Lines 62/63. Another recent study by Halicka et al (2020), published in Cortex, also found no significant evidence of visual attention bias in a large group of patients over a broad battery of tests. Similarly on line 264

Lines 65/66. This appears to be part of the rationale of the study, however it could do with rewording. Responses to visual perception tasks ARE behaviour. I think I can see what the authors mean, but they should choose different wording to express how their study differs. Perhaps something like “visual exploration of a scene” or “overt visual exploration” would be appropriate.

Lines 71-78. It’s not clear how the use of eye tracking allowed insights into attention in these studies, so the authors should make this explicit. Presumably there were differences in the participants’ looking behaviour (gaze duration etc).

Lines 318-319. It is quite likely that the absence of significant correlations could be due to the small sample size in this study.

Data should be shared as supplementary material to the paper, or uploaded to a public repository.

Reviewer #2: This study adds an interesting and novel idea for assessing pathophysiological changes in CRPS. This paper would, however, benefit greatly from some language editing and restructuring as the manner and extent of differences between healthy controls and patients discovered is quite hard to understand.

The number of participants was quite low and the variety of inciting events of CRPS was high. The effort to match controls in age and gender and the fact that only patients with CRPS type 1 in the upper limb with the same dominant hand have been included aids the quality of the study. Nevertheless the low number of participants should clearly be stated as a limitation.

I therefore suggest adding „a preliminary study“ to the titel.

Introduction:

lls 64: „Furthermore, most studies of spatial bias in CRPS have assessed responses to the task…“

The meaning of this sentence is unclear. Please clarify.

Material and methods

Please state, which limb was affected. The affected hand being the dominant one could influence the results.

The section Pain and other clinical evaluations requires some more structure.

Especially the part regarding the HADS is not clear. Please explain how the cut-off can be 10/11, when 21 points can be reached on each subscale.

ll 108: „Cut-off for chronic pain was 10 points in Japanese“

The meaning of this sentence is unclear. Please clarify.

lls 142: Please define fixation count and fixation duration using complete sentences.

Figure 1: Please explain how „nonspecific neck shoulder pain“ is an inciting event for CRPS.

There is quite the variety in inciting events (contusion of upper limb to blood sampling). Please add this to the limitations!

Results

The results section in general requires more structure. It is not always clear which side is being referred to. For example: Is „the painful side“ referring to the side of the person in the video corresponding tot he painful side of the patient? This is especially important as you refer to this in your discussion, but it is hard to understand the parts of the results section you are referring to.

lls 199: „In contrast, CRPS patients showed significant differences in FDs on the face and both forearms, and in FCs on the face and nonpainful forearm (p< 0.05, Table 2), but not in FDs and FCs on the face and body.“

Please state how these were different. Were they higher or lower?

lls 210: „Other clinical parameters, such as pain intensity, disability, disease duration, and psychological state (PCS, HADS) did not show any association with the eye-movement data.“

Does this mean no correlation?

Discussion:

Could the change of visual attention be explained by a painful disorder of the upper extremity in general and therefore not be specific to CRPS? Please discuss this. An additional control group with a non-CRPS painful disorder oft he upper extremity could have been added. Please add this tot he limitations.

„However, this study showed that the visual attention on faces tended to be lower in CRPS patients than in healthy controls. In particular, this tendency was found in patients with low body cognitive distortions.“ (ll 255)

How do you explain this. Please discuss.

Please discuss, why the forearm seemed to play a bigger role than the hand. Was the forearm affected in a majority of your patients?

The number of participants was very low and as you stated eye movement varies greately between individuals.

How much variation was there between healthy individuals?

You did not find the same association between neglect-like symptoms (or the eye-tracking parameters you hypothesise might be assosciated with those) as Sumitani et al. Please discuss why this might be.

Please discuss possible social disadvantages you state might occure for CRPS patients. Are there previous studies reporting those?

Minor

ll 48: superfluous space after hyperalgesia

ll 79: „also would have not only changes in special attention“ -> Do you mean „spacial“?

ll 110: „the exaggeration of pain“ -> Please consider rephrasing.

ll 110: superfluous komma after anxiety

ll 123: please explain the abbreviation TFT

ll 126 „Eye-tracking glasses are equipped…“ Please stay consistent using past tense.

lls 241: „Note that healthy control gazed at the face in Video 1 (Fig. 6E) and not gazed

at the hand before being touched in Video 2 (Fig. 6F).“ -> did not gaze

lls 272: „and visual attention for face“ for the face

6. PLOS authors have the option to publish the peer review history of their article (what does this mean?). If published, this will include your full peer review and any attached files.

Reviewer #1: **Yes: **Dr Janet H Bultitude

Reviewer #2: **Yes: **Ralf Baron

---

## [Author Response · Author response to Decision Letter 0]

19 Nov 2020

Dear Editor in chief and Reviewers

Thank you for all the helpful suggestions.

We have corrected it according to your suggestions and will resubmit a revised version.

Also we have uploaded additional supplement file this time.

Takahiro Ushida 

Reviewer Comments and Response

Reviewer #1: 

This is an interesting study. I have only a few minor comments

1. Lines 62/63. Another recent study by Halicka et al (2020), published in Cortex, also found no significant evidence of visual attention bias in a large group of patients over a broad battery of tests. Similarly on line 264

Response：Thank you for pointing this out to us. We have changed the following sentence on line 62. In addition, we have added reference number 8 to line 283 as follows.

“However, Christophe L (2016) 8 and Halicka M (2020) 9 reported…“

2. Lines 65/66. This appears to be part of the rationale of the study, however it could do with rewording. Responses to visual perception tasks ARE behaviour. I think I can see what the authors mean, but they should choose different wording to express how their study differs. Perhaps something like “visual exploration of a scene” or “overt visual exploration” would be appropriate.

Response: Thank you for pointing this out to us. We have reworded this section by changing “visual attentional behavior” to “overt visual exploration” on lines 66-67.

3. Lines 71-78. It’s not clear how the use of eye tracking allowed insights into attention in these studies, so the authors should make this explicit. Presumably there were differences in the participants’ looking behaviour (gaze duration etc).

Response: Thank you for your comments. We have added the assessment of visual attentional behavior described in the following sentence on lines 72-78.

“For example, a person with body dysmorphic disorder (BDD) paid more attention (i.e. increased fixations and dwell time) to their body parts that they thought were unattractive.12 Depressed participants paid more attention (i.e. increased fixation duration) to sad faces significantly more than healthy controls17. Chronic headache participants demonstrated a significantly greater visual attentional bias (i.e. fixation duration and the location of the initial shift in gaze) toward pain-related information (painful face images) even when other emotional stimuli (e.g. angry, happy, and neutral face images) were presented.15”

4. Lines 318-319. It is quite likely that the absence of significant correlations could be due to the small sample size in this study.

Response: Thank you for pointing this out to us. We have added the following sentence to lines 360-362.

“However, the absence of significant correlations could be due to the small sample size in this study, and therefore we thought that a study with a larger sample size is needed.”

5. Data should be shared as supplementary material to the paper, or uploaded to a public repository.

Response: We will share the data as supplementary material.

 

Reviewer #2: 

This study adds an interesting and novel idea for assessing pathophysiological changes in CRPS. This paper would, however, benefit greatly from some language editing and restructuring as the manner and extent of differences between healthy controls and patients discovered is quite hard to understand.

1. The number of participants was quite low and the variety of inciting events of CRPS was high. The effort to match controls in age and gender and the fact that only patients with CRPS type 1 in the upper limb with the same dominant hand have been included aids the quality of the study. Nevertheless the low number of participants should clearly be stated as a limitation.

I therefore suggest adding „a preliminary study“ to the titel.

Response: Thank you for your comment. We have changed the title to “Changes in visual attentional behavior in complex regional pain syndrome: A preliminary study.”

Introduction:

2. lls 64: „Furthermore, most studies of spatial bias in CRPS have assessed responses to the task…“

The meaning of this sentence is unclear. Please clarify.

Response: Thank you for pointing out the flaws in this phrase to us. We have changed this part to the following sentence on lines 65-66.

“Furthermore, most studies of spatial bias in CRPS have assessed responses to the task such as temporal order judgment (TOJ) tasks4,5, 9, 11, …”

Material and methods

3. Please state, which limb was affected. The affected hand being the dominant one could influence the results.

Response: The right hand of all of the participants in this study was the dominant hand. 

We have stated that “Both patients and healthy subjects were right-handed.” on lines 94-95.

And we have listed the affected side of the patients in Table 1.

4. The section Pain and other clinical evaluations requires some more structure.

Especially the part regarding the HADS is not clear. Please explain how the cut-off can be 10/11, when 21 points can be reached on each subscale.

Response: Thank you for your comments. We have changed the section regarding the HADS to the following sentence on line s121-124.

“In the reliability and validity study for Japanese version of HADS, the optimal cut-off point for screening for adjustment and major depressive disorders was an anxiety score of 10, and a depression score 11, with sufficient sensitivity and specificity (91.5 and 65.4%, respectively)21.”

5. ll 108: „Cut-off for chronic pain was 10 points in Japanese“

The meaning of this sentence is unclear. Please clarify.

Response: Thank you for pointing this out to us. We have changed the following sentence on lines 112-114．

“The cut-off point for distinguishing between patients with and without chronic pain was 10 points in the Japanese version of PDAS validity study19.”

6. lls 142: Please define fixation count and fixation duration using complete sentences.

Response: In accordance with your request, we have changed the following sentence on lines 150-154.

“Fixation Count (FC): The number of times a participant’s gaze fixed on the areas of Interest (AOI; defined below) such as face (the dotted area in Fig.1), affected side of the forearm, non-affected side of the forearm (forearms include hands, the shaded area in Fig.1), body area (the horizontal line area in Fig.1). Fixation Duration (FD): The total duration of a participant’s gaze on the AOI such as face, arm, or other body area during the video clip.”

7. Figure 1: Please explain how „nonspecific neck shoulder pain“ is an inciting event for CRPS.

Response: Thank you for your comment. The No. 5 patient had a long history of stiff neck and shoulders. After slept wrong, she developed severe non-specific neck pain with no major neurological/radiological findings, and developed CRPS symptoms afterwards. We have added the following sentence in Table 1.

“Non-specific severe neck pain: She had a long history of stiff neck and shoulders, after slept wrong, she developed severe non-specific neck pain with no major neurological/radiological findings, developed CRPS symptoms afterwards.”

8. There is quite the variety in inciting events (contusion of upper limb to blood sampling). Please add this to the limitations!

Response: Thank you for pointing this out to us. We have added the following sentence to lines 372-373.

“Thirdly, there were quite a few different types of inciting events (blood sampling to traffic accident) and this may have influenced the results.”

Results

9. The results section in general requires more structure. It is not always clear which side is being referred to. For example: Is „the painful side“ referring to the side of the person in the video corresponding tot he painful side of the patient? This is especially important as you refer to this in your discussion, but it is hard to understand the parts of the results section you are referring to.

Response: Thank you for pointing out the confusion caused by the wording in our previous manuscript. We have added the following sentence to lines 212-214 which we hope will clarify this matter.

“The painful and non-painful side refers to the side of the person in the video corresponding to the painful (VP) and non-painful (VNP) side of the patients.”

In addition, we have added “The left and right sides refer to the left (VL) and right (VR) sides of the person in the video” to lines 216-217; “The painful side and non-painful sides refer to the side of the person B in the video (Fig 1) corresponding to the painful and non-painful sides of the patients.” to lines 232-234; and “The right and left hands refer to the right and left hands of person B in the video (Fig 1)” to lines 243-244.

We have also changed the figure legend to reflect these.

10. lls 199: „In contrast, CRPS patients showed significant differences in FDs on the face and both forearms, and in FCs on the face and nonpainful forearm (p< 0.05, Table 2), but not in FDs and FCs on the face and body.“

Please state how these were different. Were they higher or lower?

Response: Thank you for pointing this out to us. We have changed the following sentence on lines 208-211.

“In contrast, facial FDs were significantly longer in CRPS patients than FDs on both forearms, and facial FCs were significantly higher than non-painful forearm FCs (p< 0.05, Table 2), but there was no significant difference between the face and body in FDs or FCs (p=0.664, 1.00, Table 2).”

11. lls 210: „Other clinical parameters, such as pain intensity, disability, disease duration, and psychological state (PCS, HADS) did not show any association with the eye-movement data.“

Does this mean no correlation?

Response: Yes. That means there was no correlation. To clarify this point, we have changed the previous phrase of “did not show any association” to the current “did not show significant correlations” on lines 223-224.

Discussion:

12. Could the change of visual attention be explained by a painful disorder of the upper extremity in general and therefore not be specific to CRPS? Please discuss this. An additional control group with a non-CRPS painful disorder of the upper extremity could have been added. Please add this to the limitations.

Response: Thank you for your suggestion. We have added the following sentence to lines 367-373.

“In addition, patients suffering from pain other than CRPS were not included in this study. NLS has been reported to occur in patients with other chronic pain as well as CRPS. For example, Frettlöh et al. observed NLS in 80.3% of patients with chronic upper limb pain35. Thus, changes in visual attentional behavior may not be specific to CRPS and should be compared to the visual attentional behavior of non-CRPS painful disorder of the upper extremity.”

13. „However, this study showed that the visual attention on faces tended to be lower in CRPS patients than in healthy controls. In particular, this tendency was found in patients with low body cognitive distortions.“ (ll 255)

How do you explain this. Please discuss.

Response: We have changed the following sentence on lines 276-291.

“The previous study reported that CRPS patients show attentional bias away from both the affected limb and the affected space4,5. Moreover, the degree to which attention was directed away from the affected side is related to BPDS4. The results of this study also showed a tendency for decreased attention to the affected limb in patients with high body cognitive distortions. Thus, we thought that the patient with high body cognitive distortions could have gazed at the face as well as healthy controls, because they reduced their attention on the affected limb.

On the other hand, it has also been reported that there was no visuospatial attention bias to the affected space in CRPS patients8,9. In addition, it also has been reported that CRPS patients have exaggerated somatosensory input from the affected limb, resulting in an overrepresentation of space on the affected side19. The results of the present study showed a positive correlation between BPDS and gazes at the face whereas there was a negative correlation with gazes at the painful forearm. Thus, we thought that the patient with low body cognitive distortions may have decreased visual attention on the face as a result of visual attentional biases to the affected limb. Therefore, the lower visual attention to faces in CRPS patients compared to healthy controls may be due to the inclusion of several CRPS patients with low body cognitive distortions in the subjects.” 

14. Please discuss, why the forearm seemed to play a bigger role than the hand. Was the forearm affected in a majority of your patients?

Response: The hand was included in the forearm, as all subjects in this study had CRPS symptoms more proximal than the hand. We have added “forearms include hands” to line 152. 

In addition, we did not analyze gaze behavior separately for the forearm and hand because the area was too small to analyze with the hand alone. However, as you correctly pointed out, we also think it is necessary to analyze the gaze behavior of the hand and forearm separately. Thus, we have added the following sentence to the Limitations section, lines 375-376.

“Especially, we thought it was necessary to analyze the gaze behavior on the forearm and hand separately.”

15. The number of participants was very low and as you stated eye movement varies greately between individuals.

How much variation was there between healthy individuals?

Response: As shown in Figure 2 and Table 3, there was less variability in visual attentional behavior in healthy subjects than CRPS patients. We will share the data as supplementary material.

16. You did not find the same association between neglect-like symptoms (or the eye-tracking parameters you hypothesise might be assosciated with those) as Sumitani et al. 

Please discuss why this might be.

Response: Thank you for your comments. Based on lines 306-338 of the Discussion section, we have added the following sentence to lines 339-346.

“In this way, there was no consensus as to whether visual attention in CRPS patients deviated to the painful or non-painful side. One possible reason to explain this is that there are different ways to assess visual attentional behavior. However, the results of the present study, which directly assessed visual attentional behavior using eye tracking systems, also showed these two patterns. As suggested by several previous studies2,4,5,6,11, we thought that the visual attentional biases in CRPS patients may be caused by different mechanisms: pain may have a strong influence on the visual attentional biases, and of body cognition distortion may have a strong influence on the visual attentional biases.”

17. Please discuss possible social disadvantages you state might occure for CRPS patients. Are there previous studies reporting those?

Response: There are no previous studies reporting social disadvantages for CRPS patients. There was an error in the wording of the English, so we have made the following changes to line 296.

“Therefore, we are concerned that reduced visual attention to the face in CRPS patients may lead to social disadvantage.” 

In addition, we have added the following sentence on the importance of visual attention to the face to lines 293-296.

“Facial expressions are important for interindividual communication, from which observers can quickly and easily make a number of inferences - about identity, gender, age, physical health, attractiveness, emotional state, personality traits, pain or physical pleasure, and even social status29.”

Minor

1. ll 48: superfluous space after hyperalgesia

Response: We have removed the superfluous space after hyperalgesia on line 48.

2. ll 79: „also would have not only changes in special attention“ -> Do you mean „spacial“?

Response: We have changed “special attention” to “spatial attention” on line 82.

3. ll 110: „the exaggeration of pain“ -> Please consider rephrasing.

Response: We have changed “the exaggeration of pain” to “the catastrophic thinking related to pain” on line 115.

4. ll 110: superfluous comma after anxiety

Response: We have removed the superfluous comma after anxiety on line 116.

5. ll 123: please explain the abbreviation TFT

Response: We have added “(Thin Film Transistor)” after TFT on line 131.

6. ll 126 „Eye-tracking glasses are equipped…“ Please stay consistent using past tense.

Response: We have changed it to “Eye-tracking glasses were equipped” on line 134.

7. lls 241: „Note that healthy control gazed at the face in Video 1 (Fig. 6E) and not gazed

at the hand before being touched in Video 2 (Fig. 6F).“ -> did not gaze

Response: We have changed “not gazed” to “did not gaze” on lines 256-257.

8. lls 272: „and visual attention for face“ for the face

Response: We have changed “for face” to “for the face” on line 275.

Reviewer Comments and Response

Reviewer #1: 

This is an interesting study. I have only a few minor comments

1. Lines 62/63. Another recent study by Halicka et al (2020), published in Cortex, also found no significant evidence of visual attention bias in a large group of patients over a broad battery of tests. Similarly on line 264

Response：Thank you for pointing this out to us. We have changed the following sentence on line 62. In addition, we have added reference number 8 to line 283 as follows.

“However, Christophe L (2016) 8 and Halicka M (2020) 9 reported…“

2. Lines 65/66. This appears to be part of the rationale of the study, however it could do with rewording. Responses to visual perception tasks ARE behaviour. I think I can see what the authors mean, but they should choose different wording to express how their study differs. Perhaps something like “visual exploration of a scene” or “overt visual exploration” would be appropriate.

Response: Thank you for pointing this out to us. We have reworded this section by changing “visual attentional behavior” to “overt visual exploration” on lines 66-67.

3. Lines 71-78. It’s not clear how the use of eye tracking allowed insights into attention in these studies, so the authors should make this explicit. Presumably there were differences in the participants’ looking behaviour (gaze duration etc).

Response: Thank you for your comments. We have added the assessment of visual attentional behavior described in the following sentence on lines 72-78.

“For example, a person with body dysmorphic disorder (BDD) paid more attention (i.e. increased fixations and dwell time) to their body parts that they thought were unattractive.12 Depressed participants paid more attention (i.e. increased fixation duration) to sad faces significantly more than healthy controls17. Chronic headache participants demonstrated a significantly greater visual attentional bias (i.e. fixation duration and the location of the initial shift in gaze) toward pain-related information (painful face images) even when other emotional stimuli (e.g. angry, happy, and neutral face images) were presented.15”

4. Lines 318-319. It is quite likely that the absence of significant correlations could be due to the small sample size in this study.

Response: Thank you for pointing this out to us. We have added the following sentence to lines 360-362.

“However, the absence of significant correlations could be due to the small sample size in this study, and therefore we thought that a study with a larger sample size is needed.”

5. Data should be shared as supplementary material to the paper, or uploaded to a public repository.

Response: We will share the data as supplementary material.

 

Reviewer #2: 

This study adds an interesting and novel idea for assessing pathophysiological changes in CRPS. This paper would, however, benefit greatly from some language editing and restructuring as the manner and extent of differences between healthy controls and patients discovered is quite hard to understand.

1. The number of participants was quite low and the variety of inciting events of CRPS was high. The effort to match controls in age and gender and the fact that only patients with CRPS type 1 in the upper limb with the same dominant hand have been included aids the quality of the study. Nevertheless the low number of participants should clearly be stated as a limitation.

I therefore suggest adding „a preliminary study“ to the titel.

Response: Thank you for your comment. We have changed the title to “Changes in visual attentional behavior in complex regional pain syndrome: A preliminary study.”

Introduction:

2. lls 64: „Furthermore, most studies of spatial bias in CRPS have assessed responses to the task…“

The meaning of this sentence is unclear. Please clarify.

Response: Thank you for pointing out the flaws in this phrase to us. We have changed this part to the following sentence on lines 65-66.

“Furthermore, most studies of spatial bias in CRPS have assessed responses to the task such as temporal order judgment (TOJ) tasks4,5, 9, 11, …”

Material and methods

3. Please state, which limb was affected. The affected hand being the dominant one could influence the results.

Response: The right hand of all of the participants in this study was the dominant hand. 

We have stated that “Both patients and healthy subjects were right-handed.” on lines 94-95.

And we have listed the affected side of the patients in Table 1.

4. The section Pain and other clinical evaluations requires some more structure.

Especially the part regarding the HADS is not clear. Please explain how the cut-off can be 10/11, when 21 points can be reached on each subscale.

Response: Thank you for your comments. We have changed the section regarding the HADS to the following sentence on line s121-124.

“In the reliability and validity study for Japanese version of HADS, the optimal cut-off point for screening for adjustment and major depressive disorders was an anxiety score of 10, and a depression score 11, with sufficient sensitivity and specificity (91.5 and 65.4%, respectively)21.”

5. ll 108: „Cut-off for chronic pain was 10 points in Japanese“

The meaning of this sentence is unclear. Please clarify.

Response: Thank you for pointing this out to us. We have changed the following sentence on lines 112-114．

“The cut-off point for distinguishing between patients with and without chronic pain was 10 points in the Japanese version of PDAS validity study19.”

6. lls 142: Please define fixation count and fixation duration using complete sentences.

Response: In accordance with your request, we have changed the following sentence on lines 150-154.

“Fixation Count (FC): The number of times a participant’s gaze fixed on the areas of Interest (AOI; defined below) such as face (the dotted area in Fig.1), affected side of the forearm, non-affected side of the forearm (forearms include hands, the shaded area in Fig.1), body area (the horizontal line area in Fig.1). Fixation Duration (FD): The total duration of a participant’s gaze on the AOI such as face, arm, or other body area during the video clip.”

7. Figure 1: Please explain how „nonspecific neck shoulder pain“ is an inciting event for CRPS.

Response: Thank you for your comment. The No. 5 patient had a long history of stiff neck and shoulders. After slept wrong, she developed severe non-specific neck pain with no major neurological/radiological findings, and developed CRPS symptoms afterwards. We have added the following sentence in Table 1.

“Non-specific severe neck pain: She had a long history of stiff neck and shoulders, after slept wrong, she developed severe non-specific neck pain with no major neurological/radiological findings, developed CRPS symptoms afterwards.”

8. There is quite the variety in inciting events (contusion of upper limb to blood sampling). Please add this to the limitations!

Response: Thank you for pointing this out to us. We have added the following sentence to lines 372-373.

“Thirdly, there were quite a few different types of inciting events (blood sampling to traffic accident) and this may have influenced the results.”

Results

9. The results section in general requires more structure. It is not always clear which side is being referred to. For example: Is „the painful side“ referring to the side of the person in the video corresponding tot he painful side of the patient? This is especially important as you refer to this in your discussion, but it is hard to understand the parts of the results section you are referring to.

Response: Thank you for pointing out the confusion caused by the wording in our previous manuscript. We have added the following sentence to lines 212-214 which we hope will clarify this matter.

“The painful and non-painful side refers to the side of the person in the video corresponding to the painful (VP) and non-painful (VNP) side of the patients.”

In addition, we have added “The left and right sides refer to the left (VL) and right (VR) sides of the person in the video” to lines 216-217; “The painful side and non-painful sides refer to the side of the person B in the video (Fig 1) corresponding to the painful and non-painful sides of the patients.” to lines 232-234; and “The right and left hands refer to the right and left hands of person B in the video (Fig 1)” to lines 243-244.

We have also changed the figure legend to reflect these.

10. lls 199: „In contrast, CRPS patients showed significant differences in FDs on the face and both forearms, and in FCs on the face and nonpainful forearm (p< 0.05, Table 2), but not in FDs and FCs on the face and body.“

Please state how these were different. Were they higher or lower?

Response: Thank you for pointing this out to us. We have changed the following sentence on lines 208-211.

“In contrast, facial FDs were significantly longer in CRPS patients than FDs on both forearms, and facial FCs were significantly higher than non-painful forearm FCs (p< 0.05, Table 2), but there was no significant difference between the face and body in FDs or FCs (p=0.664, 1.00, Table 2).”

11. lls 210: „Other clinical parameters, such as pain intensity, disability, disease duration, and psychological state (PCS, HADS) did not show any association with the eye-movement data.“

Does this mean no correlation?

Response: Yes. That means there was no correlation. To clarify this point, we have changed the previous phrase of “did not show any association” to the current “did not show significant correlations” on lines 223-224.

Discussion:

12. Could the change of visual attention be explained by a painful disorder of the upper extremity in general and therefore not be specific to CRPS? Please discuss this. An additional control group with a non-CRPS painful disorder of the upper extremity could have been added. Please add this to the limitations.

Response: Thank you for your suggestion. We have added the following sentence to lines 367-373.

“In addition, patients suffering from pain other than CRPS were not included in this study. NLS has been reported to occur in patients with other chronic pain as well as CRPS. For example, Frettlöh et al. observed NLS in 80.3% of patients with chronic upper limb pain35. Thus, changes in visual attentional behavior may not be specific to CRPS and should be compared to the visual attentional behavior of non-CRPS painful disorder of the upper extremity.”

13. „However, this study showed that the visual attention on faces tended to be lower in CRPS patients than in healthy controls. In particular, this tendency was found in patients with low body cognitive distortions.“ (ll 255)

How do you explain this. Please discuss.

Response: We have changed the following sentence on lines 276-291.

“The previous study reported that CRPS patients show attentional bias away from both the affected limb and the affected space4,5. Moreover, the degree to which attention was directed away from the affected side is related to BPDS4. The results of this study also showed a tendency for decreased attention to the affected limb in patients with high body cognitive distortions. Thus, we thought that the patient with high body cognitive distortions could have gazed at the face as well as healthy controls, because they reduced their attention on the affected limb.

On the other hand, it has also been reported that there was no visuospatial attention bias to the affected space in CRPS patients8,9. In addition, it also has been reported that CRPS patients have exaggerated somatosensory input from the affected limb, resulting in an overrepresentation of space on the affected side19. The results of the present study showed a positive correlation between BPDS and gazes at the face whereas there was a negative correlation with gazes at the painful forearm. Thus, we thought that the patient with low body cognitive distortions may have decreased visual attention on the face as a result of visual attentional biases to the affected limb. Therefore, the lower visual attention to faces in CRPS patients compared to healthy controls may be due to the inclusion of several CRPS patients with low body cognitive distortions in the subjects.” 

14. Please discuss, why the forearm seemed to play a bigger role than the hand. Was the forearm affected in a majority of your patients?

Response: The hand was included in the forearm, as all subjects in this study had CRPS symptoms more proximal than the hand. We have added “forearms include hands” to line 152. 

In addition, we did not analyze gaze behavior separately for the forearm and hand because the area was too small to analyze with the hand alone. However, as you correctly pointed out, we also think it is necessary to analyze the gaze behavior of the hand and forearm separately. Thus, we have added the following sentence to the Limitations section, lines 375-376.

“Especially, we thought it was necessary to analyze the gaze behavior on the forearm and hand separately.”

15. The number of participants was very low and as you stated eye movement varies greately between individuals.

How much variation was there between healthy individuals?

Response: As shown in Figure 2 and Table 3, there was less variability in visual attentional behavior in healthy subjects than CRPS patients. We will share the data as supplementary material.

16. You did not find the same association between neglect-like symptoms (or the eye-tracking parameters you hypothesise might be assosciated with those) as Sumitani et al. 

Please discuss why this might be.

Response: Thank you for your comments. Based on lines 306-338 of the Discussion section, we have added the following sentence to lines 339-346.

“In this way, there was no consensus as to whether visual attention in CRPS patients deviated to the painful or non-painful side. One possible reason to explain this is that there are different ways to assess visual attentional behavior. However, the results of the present study, which directly assessed visual attentional behavior using eye tracking systems, also showed these two patterns. As suggested by several previous studies2,4,5,6,11, we thought that the visual attentional biases in CRPS patients may be caused by different mechanisms: pain may have a strong influence on the visual attentional biases, and of body cognition distortion may have a strong influence on the visual attentional biases.”

17. Please discuss possible social disadvantages you state might occure for CRPS patients. Are there previous studies reporting those?

Response: There are no previous studies reporting social disadvantages for CRPS patients. There was an error in the wording of the English, so we have made the following changes to line 296.

“Therefore, we are concerned that reduced visual attention to the face in CRPS patients may lead to social disadvantage.” 

In addition, we have added the following sentence on the importance of visual attention to the face to lines 293-296.

“Facial expressions are important for interindividual communication, from which observers can quickly and easily make a number of inferences - about identity, gender, age, physical health, attractiveness, emotional state, personality traits, pain or physical pleasure, and even social status29.”

Minor

1. ll 48: superfluous space after hyperalgesia

Response: We have removed the superfluous space after hyperalgesia on line 48.

2. ll 79: „also would have not only changes in special attention“ -> Do you mean „spacial“?

Response: We have changed “special attention” to “spatial attention” on line 82.

3. ll 110: „the exaggeration of pain“ -> Please consider rephrasing.

Response: We have changed “the exaggeration of pain” to “the catastrophic thinking related to pain” on line 115.

4. ll 110: superfluous comma after anxiety

Response: We have removed the superfluous comma after anxiety on line 116.

5. ll 123: please explain the abbreviation TFT

Response: We have added “(Thin Film Transistor)” after TFT on line 131.

6. ll 126 „Eye-tracking glasses are equipped…“ Please stay consistent using past tense.

Response: We have changed it to “Eye-tracking glasses were equipped” on line 134.

7. lls 241: „Note that healthy control gazed at the face in Video 1 (Fig. 6E) and not gazed

at the hand before being touched in Video 2 (Fig. 6F).“ -> did not gaze

Response: We have changed “not gazed” to “did not gaze” on lines 256-257.

8. lls 272: „and visual attention for face“ for the face

Response: We have changed “for face” to “for the face” on line 275.

---

## [Decision Letter · Decision Letter 1]

19 Jan 2021

PONE-D-20-25338R1

Changes in visual attentional behavior in complex regional pain syndrome: A preliminary study

PLOS ONE

Dear Dr. Ushida,

Thank you for submitting your manuscript to PLOS ONE. It has been looked over by the reviewers and they are broadly satisfied that their comments have been appropriately addressed. As such, I would like to think that we may be close to accepting the manuscript. However, you will see that (alongside a couple of minor points) Reviewer 2 still feels that the Discussion could be better structured. I am in agreement here - the section would benefit from a clearer narrative and, whilst I am usually unsure about the utility of subheadings, it may be that the inclusion of some will help you to organise the themes in a more approachable fashion. I also echo the recommendation that the manuscript is thoroughly reviewed for correct English - this is the last opportunity for the document to be fully ratified for language, and so a final proofing would be advised.   

We invite you to submit a revised version of the manuscript that addresses these points. Upon receipt, I shall then decide whether the manuscript can be accepted as is, or whether another review round is appropriate. 

We look forward to receiving your revised manuscript.

Kind regards,

Alastair D. Smith

Academic Editor

PLOS ONE

Reviewers' comments:

Reviewer's Responses to Questions

**Comments to the Author**

1. If the authors have adequately addressed your comments raised in a previous round of review and you feel that this manuscript is now acceptable for publication, you may indicate that here to bypass the “Comments to the Author” section, enter your conflict of interest statement in the “Confidential to Editor” section, and submit your "Accept" recommendation.

Reviewer #1: All comments have been addressed

Reviewer #2: (No Response)

2. Is the manuscript technically sound, and do the data support the conclusions?

Reviewer #1: Yes

Reviewer #2: Yes

3. Has the statistical analysis been performed appropriately and rigorously? 

Reviewer #1: Yes

Reviewer #2: Yes

4. Have the authors made all data underlying the findings in their manuscript fully available?

Reviewer #1: Yes

Reviewer #2: Yes

5. Is the manuscript presented in an intelligible fashion and written in standard English?

Reviewer #1: Yes

Reviewer #2: No

6. Review Comments to the Author

Reviewer #1: (No Response)

Reviewer #2: Thank you for considering the remarks. The manuscript improved greatly.

Please rephrase: ll73: Depressed participants paid more attention (i.e. increased fixation duration) to sad faces significantly more than healthy controls.

l 102 and 105: The sentence “All participants provided written informed consent prior to participation in this study.“ appears twice

l 273 “it was reported that visual attentional bias for faces IS reduced in depressed individuals”

Please consider further structuring of the discussion as it remains hard to read. Consider adding subheadings for added clarity.

Please check thoroughly for language errors!

7. PLOS authors have the option to publish the peer review history of their article (what does this mean?). If published, this will include your full peer review and any attached files.

Reviewer #1: No

Reviewer #2: No

---

## [Author Response · Author response to Decision Letter 1]

26 Jan 2021

Reviewer Comments and Response

Reviewer #2: 

Thank you for considering the remarks. The manuscript improved greatly.

1. Please rephrase: ll73: Depressed participants paid more attention (i.e. increased fixation duration) to sad faces significantly more than healthy controls.

Response: Thank you for pointing this out to us. We have rephrased the following sentence on lines 74-76 as follows.

“Depressed patients are more likely to gaze at sad faces than happy ones, compared with non-depressed subjects.”

2. l 102 and 105: The sentence “All participants provided written informed consent prior to participation in this study.“ appears twice

Response: Thank you for pointing this out to us. We have removed “All participants provided written informed consent prior to participation in this study.” on lines 105-106.

3. l 273 “it was reported that visual attentional bias for faces IS reduced in depressed individuals”

Response: Thank you for your suggestion. We have changed “it was reported that visual attentional bias for faces is reduced in depressed individuals” on line 274.

Please consider further structuring of the discussion as it remains hard to read. Consider adding subheadings for added clarity.

Response: In accordance with your request, we have added subheadings on lines 267, 301, 326 and 370.

Please check thoroughly for language errors!

Response: We have checked for language errors and made corrections where necessary. The final native check of this paper has been done by Mr. Matthew McLaughlin.

---

## [Editor Report · Decision Letter 2]

1 Feb 2021

Changes in visual attentional behavior in complex regional pain syndrome: A preliminary study

PONE-D-20-25338R2

Dear Dr. Ushida,

Many thanks for submitting your revised manuscript. We’re pleased to inform you that it has been judged scientifically suitable for publication and will be formally accepted for publication once it meets all outstanding technical requirements.

Kind regards,

Alastair D. Smith

Academic Editor

PLOS ONE

---

## [Editor Report · Acceptance letter]

5 Feb 2021

PONE-D-20-25338R2 

Changes in visual attentional behavior in complex regional pain syndrome: A preliminary study 

Dear Dr. Ushida:

I'm pleased to inform you that your manuscript has been deemed suitable for publication in PLOS ONE. Congratulations! Your manuscript is now with our production department. 

Kind regards, 

on behalf of

Dr Alastair Smith 

Academic Editor

PLOS ONE